# Exploring prompting for dialectical machine translation: a focus on north Jordanian Arabic



Rasha Obeidat[1], Luay Alawneh[2] and Yara Al-Harahsheh[3]

[1] Department of Computer Science, Jordan University of Science and Technology, Irbid, Jordan
[2] Department of Software Engineering, Jordan University of Science and Technology, Irbid, Jordan
[3] Department of Data Science and Artificial Intelligence, AlHussein Technical University, Amman, Jordan

## ABSTRACT

Dialectal variations are common across many languages, and dialectical machine translation to the standard form of the language or other languages is crucial for effective communication with speakers of these dialects. Prompting Large Language Models (LLMs) for Machine Translation (MT) has gained popularity. However, its efficacy for dialectical MT, particularly in comparison to fine-tuning, remains underexplored, especially for regional dialects that lack parallel training and evaluation data. This study presents a new parallel dataset between Modern Standard Arabic and the Irbid dialect, the largest city in northern Jordan, specifically within the travel domain. This dataset, an extension of the MADAR multi-dialect *corpus*, comprises 12,000 entries translated by native speakers of the Irbid dialect. We also describe the guidelines and evaluation process employed to collect this dataset and present several analyses within this article. Additionally, we investigate the effectiveness of prompting LLMs, particularly GPT-4o-mini, in performing MT under zero-shot and few-shot learning settings. We compare these methods to fine-tuning approaches. This includes the use of dialect-tolerant prompts and constraints. We compare these methods to fine-tuning approaches. Results indicate that prompting, particularly few-shot learning with an optimal number of exemplars, consistently outperforms fine-tuning in our tests. Utilizing several versions of T5 and mBART50 for fine-tuning, we compared their performance with that of GPT-4o-mini, which was employed for prompting. The comparative analysis reveals a notable improvement margin, with Bilingual Evaluation Understudy (BLEU), Crosslingual Optimized Metric for Evaluation of Translation (COMET), and Recall-Oriented Understudy for Gisting Evaluation–Longest Common Subsequence (ROUGE-L) scores surpassing those of the best fine-tuned model by margins of 11.89, 0.2476, and 1.18, respectively. These findings underscore the potential of Few-shot Prompting (FSP) in effectively addressing dialectical MT challenges.

Corresponding author
Rasha Obeidat,
rmobeidat@just.edu.jo

## INTRODUCTION

Arabic, with its diverse dialects, plays a central role in the daily communication of millions across the Arab world. While Modern Standard Arabic (MSA) serves as the standardized language in formal contexts—such as education, media, and official communications—dialects dominate informal communication, particularly on digital platforms. Despite MSA's unifying role, significant linguistic differences among dialects often lead to misunderstandings between speakers from different regions. This highlights the need for specialized linguistic resources and computational models that can process and translate these informal language forms. Effective Machine Translation (MT) between dialects and the mother tongue of a society, as well as between dialects and major languages like English, is crucial for enhancing mutual understanding and supporting applications in media, business, and tourism. Addressing this need requires the development of parallel Dialect—MSA and Dialect—Dialect corpora to enable robust translation systems.

Arabic encompasses an extensive array of dialects that showcase remarkable diversity across regions, countries, and even neighboring cities. These dialects are broadly categorized as Levantine, Gulf, Egyptian, North African, Iraqi, and Yemeni (*Zbib et al., 2012*). Although translation resources exist for these major dialects into MSA and English (*Nagoudi, Elmadany & Abdul-Mageed, 2022b*; *Cettolo et al., 2017*; *Takezawa et al., 2007*), resources for dialectical Arabic remain scarce compared to the well-established corpora for MSA and other languages. Moreover, existing efforts primarily target major dialect categories (*Alzamzami & Saddik, 2023*; *Meftouh et al., 2015*), with a limited focus on country-specific (*Jarrar et al., 2017*; *Kwaik et al., 2018*) or city-specific dialects. Yet, significant distinctions exist even between dialects within the same country, at phonological, lexical, morphological, and syntactic levels (*Watson, 2012*). Resources for translating city-level dialects remain particularly scarce, making projects like MADAR (*Bouamor et al., 2018*) especially valuable for addressing this gap.

MADAR is a large parallel *corpus* covering 25 Arabic city dialects within the travel domain. Two thousand sentences from the MSA-English Basic Travel Expression Corpus (BTEC) dataset (*Takezawa et al., 2007*) were translated into these 25 dialects, forming the subset referred to as CORPUS-25. Additionally, MADAR includes CORPUS-5, comprising 10,000 sentences translated into the dialects of five cities: Beirut, Doha, Tunis, Cairo, and Rabat[1]. While MADAR provides broad coverage of Arabic city dialects, it lacks comprehensive representation of many subregional and city-specific dialects, particularly within Jordan. Aside from Amman and Salt, the corresponding subsets in CORPUS-25 are too small to support training modern language models, and other areas in Jordan remain underrepresented. Furthermore, existing subsets primarily reflect cities from the country's central region, overlooking the northern and southern cities, whose dialects differ syntactically from those of central Jordan.

To broaden the representation of underrepresented city dialects in Jordan, we have focused on Irbid, the country's largest city in the northern part. We introduce IrbidDial, a parallel (Irbid dialect)-MSA dataset that includes MSA sentences from 12,000 MSA-English sentence pairs originally sourced from BETC and used in MADAR.

[1] In a recent release of MADAR, these subsets are referred to as CORPUS-26 and CORPUS-6 after including MSA sentences.

These sentences have been manually translated into the Irbid dialect. To support cross-dialect MT research, we intentionally translated the same sentences utilized in the MADAR dataset, effectively positioning our dataset as an external extension of MADAR. Moreover, since the MSA sentences in MADAR were initially derived from BETC's MSA-English parallel sentences, this implies the IrbidDial sentences in our dataset also have corresponding English translations. To authentically capture the linguistic diversity of Irbid, eight native speakers from different districts in Irbid, maintaining an equal gender ratio, undertook the task of translating these sentences into their respective local dialects, thereby genuinely distinguishing the Irbid dialect from other cities in Jordan. Additionally, we developed a dictionary of 2,155 term pairs (Irbid dialect)-MSA intended to support future research on this dataset.

Recent studies have demonstrated that prompting Large Language Models (LLMs) with minimal data can achieve comparable or even surpassing specialized systems across various Natural Language Processing (NLP) tasks, including MT (*Zhang et al., 2023*; *Vilar et al., 2023*; *Moslem et al., 2023*). However, LLMs exhibit strong capabilities in translating high-resource languages, such as English and French. Their performance in translating lower-resource languages, by contrast, tends to be less robust (*Huang et al., 2023*). Moreover, the effectiveness of LLMs in translating dialects, particularly within the context of the Arabic language, remains underexplored. This study aims to contribute to filling this gap by leveraging GPT-4o-min dialect translation through Zero-shot Prompting (ZSP) and Few-shot Prompting (FSP), utilizing a range of tailored prompting templates and constraints, some of which are specifically designed to be dialect-focused. Additionally, the performance of prompting is compared against several LM-based MT architectures fine-tuned using the training set of the IrbidDial dataset. We experimentally demonstrate that prompting, especially when incorporating parallel (Irbid dialect)-MSA examples, consistently and marginally outperforms fine-tuning while reducing the dependency on extensive training datasets. The contributions of this work can be summarized as follows:

- **IrbidDial dataset:** We present the IrbidDial dataset, comprising 12,000 MSA sentences, manually translated into the Irbid dialect by native speakers. The MSA sentences were originally sourced from the BTIC MSA-English parallel dataset and later utilized in the MADAR project. This dataset aims to enhance the representation of the Irbid dialect—one of the primary city dialects in Jordan—within existing linguistic resources.
- **IrbidDial-MSA dictionary:** In addition to the dataset, we developed an IrbidDial dictionary containing 2,155 term pairs that map the unique expressions and vocabulary of the Irbid dialect to their corresponding MSA terms. This dictionary aims to support further research and utilization of the IrbidDial Dataset.
- **Prompting and fine-tuning results:** This study establishes baseline results by extensively prompting GPT-4o-mini for dialect MT using various prompting templates. These templates include dialect-focused prompting under both zero-shot and few-shot settings. The outcomes from the prompting methods are compared to the results of fine-tuning multiple language models commonly used for MT.

## BACKGROUND AND RELATED WORK

This section overviews parallel datasets and the primary approaches proposed for Arabic dialects MT.

### Existing parallel corpora for dialectical Arabic MT

Dialectal variations are prominent features of many languages, including Arabic. Since people frequently use dialects in daily communication, especially on social media platforms, creating parallel dialectical corpora for various languages has gained increasing attention. For instance, datasets for Spanish (*Chiruzzo et al., 2020*; *Gutierrez-Vasques, Sierra & Pompa, 2016*), Swiss German (*Dogan-Schönberger, Mäder & Hofmann, 2021*), and Vietnamese (*Le & Luu, 2023*) dialects have been proposed. These corpora typically contain parallel sentences in the dialectical and standard forms of the language or dialect-to-English sentence pairs, primarily aiming to serve MT.

Arabic has attracted significant interest due to its wide range of dialects and subdialects, which vary greatly across regions. Like other languages with dialects, the existing dialectic Arabic parallel corpora primarily focus on translating the Arabic dialects into MSA or English. While many of these corpora target the main dialect categories (*e.g.*, Levantine, Gulf, Egyptian), others aim for broader coverage by targeting finer location granularity. Besides, various neural MT models have been utilized to validate these datasets' quality and establish benchmark performance. Among the notable efforts that targeted coarse-grained dialect categories, *Zbib et al. (2012)* utilized crowdsourcing to construct Arabic-dialect Parallel Text (APT), a Levantine-English and Egyptian-English parallel *corpus* comprising 138,000 Levantine-English sentences, and 38,000 Egyptian-English sentences originally gathered from Arabic weblogs. *Alzamzami & Saddik (2023)* created OSN-MDAD, the Online Social Network-based Multidialect Arabic Dataset (OSN-MDAD), which translates English tweets into four Arabic dialects: Gulf, Yemeni, Iraqi, and Levantine. Dial2MSA (*Mubarak, 2018*) is a parallel *corpus* encompassing 6,000 tweets from four Arabic dialects: Egyptian, Levantine, Gulf, and Maghrebi. Each tweet is translated into MSA by native speakers of the respective dialects. *Krubiński et al. (2023)* present a large-scale parallel *corpus* of 120,600 sentences sampled from OpenSubtitles-v2018 (*Lison, Tiedemann & Kouylekov, 2018*), originally translated into MSA and several Indo-European languages.

Focusing on country-specific dialects, certain datasets have been developed to capture the unique linguistic characteristics of Arabic dialects within individual countries. These efforts ensure that the translation systems are more culturally and linguistically relevant for specific regions, though comprehensive examples in this category are still relatively sparse. *Bouamor et al. (2018)* introduced MDPC, a multi-dialectal parallel *Corpus* comprising 2,000 sentences in MSA, Egyptian, Tunisian, Jordanian, Palestinian, and Syrian Arabic, as well as English, all translated from Egyptian sentences. Similarly, Parallel Arabic Dialect Corpus (PADIC) (*Meftouh et al., 2015*) is a parallel *corpus* comprising approximately 6,400 sentences in MSA, as well as Algerian, Moroccan, Tunisian, Syrian, and Palestinian dialects, alongside translations in French and English. The authors further translated these sentences into North Levantine dialects. *Al-Ibrahim & Duwairi (2020)* manually created two dialectical Jordanian-MSA datasets. The first dataset, W2W, is used for word-to-word

**Table 1 Overview of existing Arabic dialect parallel corpora, including our dataset.**

| Dataset | Dialect(s) | Language pair | Size | Notes |
|---|---|---|---|---|
| APT (*Zbib et al., 2012*) | Levantine, Egyptian | Dialect–English | 176,000 | Crowdsourced; Arabic weblogs |
| OSN-MDAD (*Alzamzami & Saddik, 2023*) | Gulf, Yemeni, Iraqi, Levantine | Dialect–English | 10,000+ | Social media (tweets) |
| Dial2MSA (*Mubarak, 2018*) | Egyptian, Levantine, Gulf, Maghrebi | Dialect–MSA | 6,000 | Tweets; MSA translations by native speakers |
| OpenSubtitles (*Krubiński et al., 2023*) | MSA | MSA–Multiple Indo-European Languages | 120,600 | Movie subtitles; includes MSA only |
| MDPC (*Bouamor et al., 2018*) | Egyptian, Tunisian, Jordanian, Palestinian, Syrian | Dialect–MSA/English | 2,000 | Country-level dialects; multi-domain |
| PADIC (*Meftouh et al., 2015*) | Algerian, Moroccan, Tunisian, Syrian, Palestinian | Dialect–MSA/French/English | 6,400 | Includes translations to North Levantine |
| W2W (*Al-Ibrahim & Duwairi, 2020*) | Jordanian | Word–Word | 24,200 words | Word-level dictionary resource |
| Seq2Seq dataset (*Al-Ibrahim & Duwairi, 2020*) | Jordanian | MSA–Jordanian | 500 sentences | Sentence-level; manually created |
| Tunisian-MSA (*Kchaou, Boujelbane & Belguith, 2020*) | Tunisian | Dialect–MSA | – | Data augmentation used |
| MADAR CORPUS-25 (*Bouamor et al., 2018*) | 25 City Dialects (incl. Amman) | Dialect–MSA | ~12,000 | Travel domain; city-level focus |
| MADAR CORPUS-5 (*Bouamor et al., 2018*) | Five City Dialects | Dialect–MSA | 10,000 | Selected cities only; no Jordanian cities |
| Our dataset | Irbid (North Jordan) | (Irbid dialect)–MSA | 12,000 | Largest dataset for northern Jordan dialect; includes 2,155-entry MSA-Irbid dictionary |

translation and contains 24,200 words. The second dataset, the seq2seq dataset, is used for sentence-level translation and comprises 500 sentences. *Kchaou, Boujelbane & Belguith (2020)* presented a Tunisian-MSA parallel *corpus* created using a data augmentation technique.

Moving towards an even more fine-grained dialect level that targets specific cities, *Bouamor et al. (2018)* created MADAR, a large parallel *corpus* of 25 Arabic city dialects in the travel domain. In this project, 2,000 BTEC sentences were translated into 25 city dialects, including the Amman dialect, referred to as CORPUS-25. Additionally, MADAR translated 10,000 sentences into the dialects of five selected cities: Beirut, Doha, Tunis, Cairo, and Rabat, and named this subset CORPUS-5. Table 1 summarizes the main Arabic dialect parallel corpora reviewed in this section, highlighting their coverage, size, and focus.

Notably, none of the cities covered in CORPUS-5 are in Jordan; Amman and Salt are the only Jordanian city-level dialects included in CORPUS-25. These sets are too small to train modern language models for MT and mainly represent cities from the central part of the country, overlooking the northern and southern cities that diverge significantly from central dialects. This omission highlights the need for expanded

research and dataset development to encompass a broader range of Jordanian dialects, thereby ensuring a more comprehensive representation of the country's linguistic diversity.

To address this gap, we introduce a dedicated parallel *corpus* focused on the dialect of Irbid, the main city in northern Jordan. Specifically, we translated 12,000 sentences from the MADAR CORPUS-5 and CORPUS-25 into the Irbid dialect and created a dictionary comprising 2,155 MSA-Irbid terms. A full description of this dataset is provided in the dataset description section of the article. This resource complements existing efforts by providing data for a significant yet previously underrepresented regional dialect, thereby enhancing the breadth and utility of Arabic dialectal corpora. To the best of our knowledge, our dataset is the largest dialectical parallel dataset and the only resource covering the north-of-Jordan dialect.

## Dialectical Arabic MT approaches

The increase in research on automatic dialectal MT has been fueled by the creation of parallel dialectal corpora and lexicons (*Bouamor et al., 2018*). Studies in Arabic language translation have taken diverse directions; some focus on translating between MSA and various Arabic dialects, while others aim to translate Arabic dialects into English. Additionally, the methodologies applied in this research are diverse, ranging from statistical approaches to sequence-to-sequence (seq2seq) models up to advanced language model-based methods. For instance, *Zbib et al. (2012)* utilized a phrase-based hierarchical translation approach that employed GIZA++ (*Och & Ney, 2003*) for sentence alignment and a log-linear model that incorporated multiple features as a decoder. *Sajjad, Darwish & Belinkov (2013)* introduced a transformational model that utilizes MSA as a pivot language for translating dialectal Egyptian Arabic into English. Similarly, *Sawaf (2010)* and *Salloum & Habash (2013)* also employed MSA as a pivot for translating dialects into English. Their approach involved morphological normalization of dialects to MSA prior to translation into English using a Statistical MT system.

Transitioning to Neural MT (NMT) systems, encoder-decoder architectures emerged as the initial prominent neural techniques used for MT. *Guellil, Azouaou & Abbas (2017)* utilized a vanilla Recurrent Neural Network (RNN) encoder-decoder model for translating Algerian Arabic written in a mixture of Arabizi and Arabic characters into MSA. *Al-Ibrahim & Duwairi (2020)* employed a multi-layer RNN encoder-decoder to translate text from the Jordanian dialect to MSA using the seq2seq dataset. *Baniata, Park & Park (2018)* experimented with an RNN encoder-decoder to translate Levantine and Maghrebi dialects into MSA. *Tawfik et al. (2019)* combined morphology with dialectal Arabic word segmentation and an RNN encoder-decoder model with an attention mechanism to translate Egyptian, Levantine and Gulf dialects to English.

To address the issue of parallel data scarcity, several studies have proposed using transfer learning and back translation approaches to enhance the performance of MT systems. For example, *Farhan et al. (2020)* developed a dialectal Attentional seq2seq model that leverages similarities among dialects through common words to translate dialects not

represented in the training data. Additionally, *Nagoudi, Elmadany & Abdul-Mageed (2021)* employed a Seq2Seq transformer MT model fine-tuned on data from various Arabic dialects to improve the performance of translating code-mixed MSA and Egyptian Arabic into English. *Ko et al. (2021)* implemented a combination of denoising autoencoding, back-translation, and adversarial objectives to utilize monolingual data and adapt NMT systems in multiple low-resource settings, including Arabic dialects. *Sajjad et al. (2020)* trained transformer-based seq2seq models to translate Arabic dialects to English using various training settings, including fine-tuning, back-translation, and data augmentation. They evaluated their models using AraBench, an evaluation suite that consolidates parallel sentences from existing dialectal Arabic-English resources across various genres. *Kchaou, Boujelbane & Hadrich (2023)* trained several transformer-based and RNN seq2seq models for translating Tunisian dialects into MSA using datasets collected from existing resources and enhanced with various data augmentation techniques.

As the field of MT has progressed, the focus has gradually shifted from traditional NMT systems to the more advanced LMs, which offer superior capabilities and broader language coverage. For dialectical Arabic MT, *Nagoudi, Elmadany & Abdul-Mageed (2022a)* pretrained three Arabic T5-based models and evaluated them for several text generation tasks, including MSA-dialect MT. *Fares (2024)* released the AraT5-MSAizer model, a fine-tuned transformer-based encoder-decoder model to translate dialectal Arabic into MSA. *Alahmari (2024)* fine-tuned five Arabic T5 models, namely ArabicT5, AraT5 base, AraT5v2-base-1024, AraT5-MSA-Small and AraT5-MSA-Base, to translate five Arabic dialects (Gulf, Egyptian, Levantine, Iraqi, and Maghrebi) into MSA.

Recent research has shown that prompting LLM with customized textual prompts can achieve impressive performance in MT across various tasks, delivering results comparable to traditional methods without extensive retraining (*Bawden & Yvon, 2023*; *Moslem et al., 2023*). For example, *Brown et al. (2020)* demonstrated that manually designed prompts could enable GPT-3 to perform a wide range of tasks, including MT, with robust performance. Similarly, *Vilar et al. (2023)* and *Zhang, Haddow & Birch (2023)* investigated various prompting templates and examined different example selection techniques for few-shot learning. *Moslem et al. (2023)* utilizes few-shot learning to improve real-time adaptive MT across five diverse language pairs: English-to-Arabic, English-to-Spanish, English-to-Chinese, English-to-French and English-to-Kinyarwanda. However, most of these studies have focused on prompting language models for translating standard languages, leaving a significant gap in the literature regarding the use of language models for dialect translation, particularly for Arabic. This research addresses a gap by exploring the translation of Irbid dialect to MSA, offering initial insights into this underexplored area.

## THE IRBID-TO-MSA PARALLEL DATASET

We developed the IrbidDial-MSA parallel dataset by manually translating 12,000 MSA sentences into the Irbid city dialect. The source MSA sentences were obtained from the

MADAR *corpus*: specifically, 10,000 sentences from CORPUS-5-MSA and 2,000 sentences from CORPUS-25-MSA. We refer to the 10,000 translated sentence pairs, denoted as IrbidDial-10k, which serve as our training set in the experiments. The remaining 2,000 sentence pairs are referred to as IrbidDial-2k, which we split equally into a development set (1,000 examples) and a test set (1,000 examples).

To ensure accurate and representative translations, we recruited eight native speakers of the Irbid dialect, carefully selected from six different districts within Irbid city: Bani Kenanah, Ar-Ramtha, Al-Mazar al-Shamali, Al-Wastiyah, Bani Obeid, and Qasabit Irbid. This selection aimed to capture both rural and urban dialectal variations, with a deliberate focus on incorporating features of rural dialects. These translators were the most critical resource in ensuring dialectal authenticity. All translators were university students or recent graduates from Jordan University of Science and Technology and other local universities, specializing in fields such as information technology, engineering, medicine, law, and Arabic language. Both male and female translators participated, further contributing to the linguistic diversity of the dataset.

Before participation, all translators underwent a brief interview to assess their proficiency in MSA. This step ensured their ability to understand and translate the source sentences accurately. All selected translators demonstrated a solid ability to comprehend the MSA sentences, especially given that these sentences were written in moderate-level Standard Arabic, typically encountered in the travel domain, which aligns with the school-level Arabic proficiency commonly attained by Jordanian students.

Each translator was provided with the MSA sentences and instructed to translate them into their native dialect as naturally and authentically as possible. We selected MSA as the source language to simplify the translation process and allow participation of native speakers without requiring proficiency in English or French. In Jordan, dialects are broadly categorized into urban, rural (Fallahi), and Bedouin varieties (*Mashaqba et al., 2023*). The urban dialects of cities like Irbid and Amman are relatively similar, whereas rural dialects exhibit significant variation (*Almhairat, 2015*). Additionally, sociolinguistic studies indicate that women tend to prefer using civil (urban) dialects due to their perceived prestige (*Al-Tamimi, 2001*). To ensure that our dataset differs from the MADAR Amman *corpus* and better reflects Irbid's linguistic diversity, we ensured gender balance among translators and encouraged the inclusion of rural dialect features wherever possible.

These MSA sentences in MADAR that we translated to Irbid dialect are originally obtained from the Basic BTEC, which provides each sentence in English, French, and MSA (*Takezawa et al., 2007*)[2]. We selected the MSA sentences from MADAR primarily because they have already been translated into various Arabic dialects, as well as English and French. This wide range of existing translations enhances the potential for integrating the Irbid dialect in future cross-dialect and multilingual MT research. Our dataset retains its inherent link to its corresponding English and French translations through the BTEC and MADAR corpora, as all resources use the same MSA sentence IDs, which we also adopted in this study. A summary of the dataset's key characteristics is presented in Table 2.

---

[2] The IrbidDial *corpus* will be publicly available for research purposes, including the sentence IDs and the translations into the Irbid dialect. Due to copyright restrictions, the MSA, English, and French components will not be distributed. With the IDs, the MSA sentences can be obtained directly from the MADAR Project: https://www.madar.com. For access to the English and French *corpus*, interested parties should contact NICT (https://www.nict.go.jp/) and the U-Star Consortium (http://www.ustar-consortium.com/).

**Table 2  Summary of the IrbidDial-MSA parallel dataset.**

| Feature | Description |
|---|---|
| Source sentences | 10,000 sentences from MADAR CORPUS-5-MSA and 2,000 sentences from MADAR CORPUS-25-MSA. |
| Target dialect | Irbid city dialect (Jordan). |
| Total sentence pairs | 12,000 (Irbid dialect)-MSA sentence pairs. |
| Translation process | Manual translation performed by eight Jordanian native speakers of the Irbid dialect, selected from six districts covering both rural and urban areas. |
| Translator background | University Irbidian students or graduates from Jordan University of Science and Technology and other local universities, specializing in Information Technology, Engineering, Medicine, Law, and Arabic literature. |
| Domain | Travel. |
| Example | MSA: نعم، كل عشر دقائق |
| | Irbid Dialect: أيوه، كل عشر دقايق |
| | English (reader reference only): Yes, every ten minutes. |

## Translation guidelines

We provided the native translators with the source sentences in MSA and gave them clear instructions on how to translate the sentences by focusing solely on the explicit content of the text without making any assumptions. In addition, we provided the following set of guidelines, adapted from (*Bouamor et al., 2018*), to ensure the quality of translations and consistency across all translators:

- Translators were instructed to use Arabic script and avoid code-switching to prevent incorporating trendy or short-lived terms that may become outdated, thus ensuring the translations remain accurate and timeless. For example, to translate the MSA sentence 'انها مبهرجة جدا' ('It is very flashy'), a translation such as 'لمعتها أوفر' is not acceptable, since this word 'أوفر' is a transliteration of the English word 'over' and has only relatively recently started to be used by young people. On the other hand, translators are instructed to use well-established loanwords, such as 'بنك' (bank), since the Arabic translation of the word 'bank,' which is 'مصرف', is not widely popular.

- Punctuation marks from the source sentence should generally be preserved in the Irbid dialect translation. However, permissible omissions or additions can be made if they are to enhance the authenticity and accuracy of the translation.

- When translating idioms from MSA to the Irbid dialect, they should not be word-for-word translated. Instead, the essence of the idiom is conveyed, or a corresponding local idiom is used if one exists.

- Attention should be paid to the appropriate use of the hamza (أ, إ) forms, especially in the case of second-person pronouns. For example, the MSA forms 'أنتَ' (you singular) and 'أنتم' (you plural) typically become 'إنتَ' and 'نتوا' in the Irbid dialect. Although it is common to normalize the hamza in MT to a single form (*i.e.*, ا) *Kwaik et al. (2018)*, *Al-Ibrahim & Duwairi (2020)*, we prefer to preserve the original variations in our dataset. This approach allows users to decide whether to normalize these forms or retain their distinct usage based on their specific needs.

- Numbers written as words should be translated into words in the target dialect. For example, the Arabic word for 'ثمانية عشر' (eighteen) should be translated as 'ثمنطعشر' or 'ثمنطعش' in Irbid dialect. Numbers written as digits should remain exactly as they are in the translation.

## Translation quality control

The dataset was segmented into 24 subsets, each containing 500 sentences. Each translator was assigned to translate one subset. After each translator completed the first 100 sentences, a quality check was conducted to ensure adherence to guidelines and quality standards, and feedback was provided for improvement. A sample of 100 sentence pairs was taken from all translators' translations, and the initial overall rejection rate during the quality check was 18%. The rejected translations were mainly due to violations of one or more annotating guidelines rather than the use of dialects other than Irbid's. Some translations were inaccurate for the source MSA sentences as they added extra information or omitted existing details. We provided ongoing quality feedback to our translators to help them continually improve their translations. Translators who demonstrated high-quality work and strict adherence to standards were assigned additional work sets. As the translation process progressed, the rejection rate steadily decayed over time. Translators' turnout was initially slow but increased as they became more familiar with the guidelines. Notably, most translators exhibited a serious commitment to their assignments.

For transparency, the quality checking and final review were conducted by native speakers of the Irbid dialect who were selected using the same criteria as the translators but were not involved in the initial translation of the sentences they reviewed. Their task was strictly limited to evaluating the quality of translations and compliance with guidelines.

## Translation quality assessment

To evaluate the overall quality of the collected data after completing the data annotation, we assessed the quality of the translated data by randomly sampling 500 examples from the dataset. Each example was reviewed by a dialect speaker (referred to as the reviewing worker) who was not involved in the data annotation process. The reviewing worker was assigned one of the following categories:

- **Fully Consistent Translation:** The translation is fully consistent with what the reviewing translator would provide if they had been assigned the task
- **Acceptable but Regionally Varied:** The translation is consistent with all guidelines, but the translator identifies potential regional variation within the Irbid dialect. The translator can recognize a slightly different, yet still correct, translation that may reflect the specific district from which the original translator is.
- **Correct Dialect but Partially Deviated Translation:** The translator believes that the translation is in the correct dialect (Irbid), but the translation is incorrect due to issues such as adding extra information or omitting details, resulting in a partial deviation from the correct sentence structure.

**Table 3 Translation quality categories with corresponding percentage distribution.**

| Category | Percentage |
| --- | --- |
| Fully consistent translation | 82.8% |
| Acceptable but regionally varied | 0.4% |
| Correct dialect but partially degraded | 4.6% |
| Completely incorrect dialect or translation | 3.8% |
| Mixed dialect | 3.8% |
| Guideline violation | 4.6% |

- **Completely incorrect Dialect or translation:** The translation is either in a completely different dialect or MSA or completely incorrect.
- **Mixed Dialect:** The translation uses Irbid dialect but contains elements of MSA or other dialects.
- **Guideline Violation:** The translation is correct and uses Irbid dialect only, but violates one or more guidelines (*e.g.*, code-switching, numbers are not translated into words, punctuation marks are not preserved).

As summarized in Table 3, a significant portion of the translations are of high quality, with the majority categorized as fully consistent (82.8%) or acceptable with slight regional variations (0.4%). This reflects the overall robustness of the dataset and its alignment with dialect-specific translation guidelines. The relatively low percentages of partially deviated translations, mixed dialects, and guideline violations further showcase the quality of the dataset.

## Irbid dialect to MSA dictionary

The (Irbid dialect)-MSA dictionary is designed to facilitate the translation process from MSA to the Irbid dialect. It aligns MSA concepts with their Irbid dialect counterparts, helping users quickly grasp dialectal variations without delving into complex phonological or orthographic details, such as Conventional Orthography for Dialectal Arabic (CODA) or Corpus of Annotated Phonological and Historical Information (CAPHI) (*Habash et al., 2018*), which are slated for future development. The dictionary contains 2,155 MSA entries, each listing an MSA concept followed by one or more translations in the Irbid dialect, covering a broad spectrum of daily vocabulary and expressions. These translations may appear as single words or phrases, reflecting the diverse ways an MSA word can be conveyed in the dialect and *vice versa*. For example, the MSA concept بطء (slowly) translates to شوي شوي or عمهل in the Irbid dialect, and كيف حالك (how are you?) translates to شلونك or كيفك. The average number of Irbid dialect terms per MSA term is 1.23.

## Data analysis

**Sentence Length Distribution** The sentence length in the IrbidDial dataset ranges from 1 to 59 words. Figure 1 represents the word-based length distribution of MSA and Irbid dialect sentences in the IrbidDial-10k and IrbidDial-2k datasets, categorized into short (1–6 words), medium (7–12 words), and long (13 words or more) sentences. It is

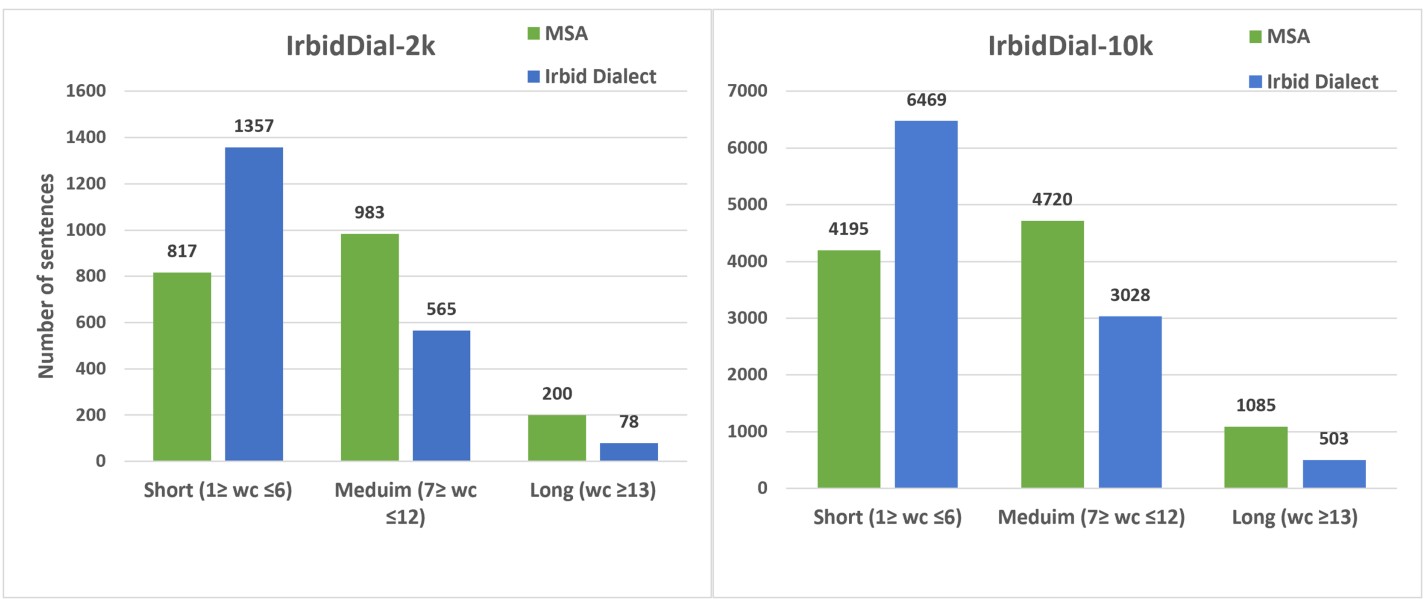

**Figure 1** Word-based MSA and Irbid dialect parallel sentence length distribution, categorized into short, medium, and long sentences.

evident that the majority of the sentences are either short or medium in length. Furthermore, the sentences in the Irbid dialect are generally shorter than their MSA equivalents. This is because, like other dialects, the Irbid dialect lacks certain grammatical structures that are required in MSA for a sentence to be considered complete. For instance, in MSA, when asking a question, an interrogative word such as 'هل' and the subordinating conjunction 'أن' are often used, while in the Irbid dialect, they are omitted because their meaning is implicitly understood. For example, the six-word MSA sentence 'هل يمكن أن آخذ بعض القهوة؟' ('Can I get some coffee?') is equivalent to the four-word sentence 'ممكن آخذ شوية قهوة؟' in the Irbid dialect, with the Irbid dialect sentence being shorter due to the omission of these formal grammatical structures.

**Word-Based Lexical Overlap Distribution** In our exploration of the linguistic commonality between the Irbid dialect and MSA, we analyzed the word-based lexical overlap between each MSA sentence and its corresponding Irbid dialect translation, calculated as the percentage of shared words between the two sentences. This reflects the degree of direct lexical similarity (syntactic overlap) between the sentence pairs. Based on this, we systematically categorized the sentences into three distinct overlap levels: low overlap (less than 35%), medium overlap (35–70%), and high overlap (above 70%). We then counted the samples in each category and presented the results in Fig. 2. Many sentence pairs exhibited low overlap (7,036 out of 12,000 pairs), highlighting the substantial lexical divergence between the Irbid dialect and MSA. This underscores the need for large-scale parallel datasets to support translation between MSA and fine-grained regional dialects like Irbid's, helping bridge the communication gap between these distinct forms of Arabic.

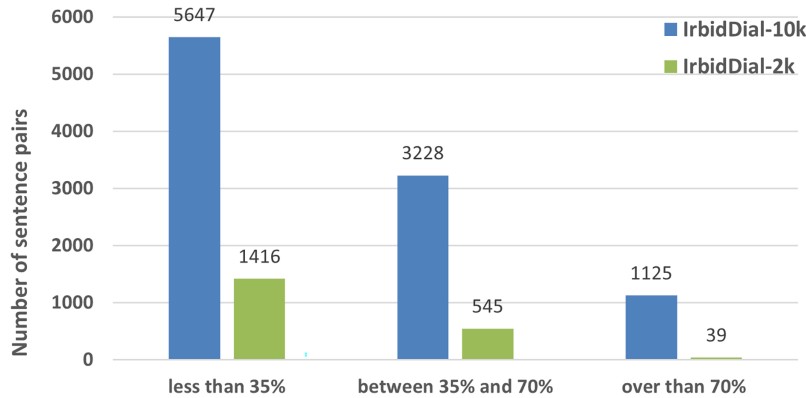

**Figure 2 Word overlap between (Irbid dialect)-MSA sentence pairs in IrbidDial-10k and IrbidDial-2k.** 

**Table 4 sample of parallel sentence in MSA, Irbid dialect and Amman Dialect (obtained from MADAR.Amman) illustrating the lexical and morphological disparities between sentences between MSA and these dialects.**

| Subset | Senetnce |
|---|---|
| MSA | هذا بارد إلى حد ما. هل يمكن أن تقوم بتسخينه؟ |
| | *haðA bArid ĂlA Hadī mA. Hal Yumkin Ân takuwm?* |
| MADAR.Amm | هاد بارد شوي. ممكن تسخنه؟ |
| | *hAd bArd šway. Mumkin tsaxnuh?* |
| Irbid dialect | هاظ بارد شوي. بتقدر تسخنه؟ |
| | *hAĎ bArid šway. btigdar tsaxnuh?* |
| English | This is a bit cold. Could you heat it up? |
| MSA | تقدم إلّ المساعدة، شكراً لك |
| | *taqadama Ălý̄ý AlmusAçadħ, šukrā lak* |
| MADAR.AMMAN | حد عم بساعدني، شكراً الك |
| | *Had çm bisAçdny,šukrā Ălak* |
| Irbid dialect | في حدا قاعد بساعدني، شكراً الك |
| | *fy HadA qaçid bisaçdny, šukrā Ălak* |
| English | Currently, assistance is being provided to me, thank you. |
| MSA | لا أدري كم من النقود يجب علّ تركه كبقشيش |
| | *lA Âdry kam min Alnuqwd yajib çalaý tarkuh kabaqšyš* |
| MADAR.AMMAN | أنا بدي أعرف كم لازم أترك بغشيش |
| | *Âna bdy Âçraf kam lazym Âtruk bagšyš* |
| IrbidDial | ما بعرف قديش لازم أترك بخشيش |
| | *mA baçraf qdyš lAzm Âtruk baxšyš* |
| English | I don't know how much money I should leave as a tip. |
| MSA | نرغب في أن نقيم أربع ليالي بدءاً من أول أغسطس |
| | *narγab fy Ân nuqym Ârbç layAly bd'āmn Âwal Âuγustus* |
| MADAR.AMMAN | احنا حابيين نضل أربع ليالي من واحد أغسطس |
| | *AHnA HAbyyn nDal Ârbç layAly min wAHd Âuγustus* |
| IrbidDial | احنا حابيين نقعد أربع ليالي اعتباراً من واحد ثمنية |
| | *AHnA HAbyyn nuqçd Ârbç layAly AçtibArnā min waHd tamanya* |
| English | We would like to stay for four nights starting from the first of August. |

## Lexical and Morphological Comparison of Irbid Dialect, MSA, and Amman Dialect

Table 4 presents a sample of parallel sentences in MSA, Irbid dialect (from the IrbidDial-2k dataset), and Amman dialect (from MADAR.Amman), illustrating lexical and morphological disparities between MSA and these dialects. For example, the MSA term 'نقيم' (stay) translates as 'نضل' in the Amman dialect and 'نقعد' in the Irbid dialect. Similarly, 'هذا' (this) is translated to 'هاد' in the Amman dialect and 'هاظ' or 'هاذ' in the Irbid dialect. Nevertheless, certain terms such as 'شوي' (a bit) and 'حابيين' (we would like to) are shared by both Irbid and Amman dialects. This not only exemplifies the diversity among the dialects within the same country but also the lexical and morphological affinities that frequently exist among dialects of nearby cities. This aligns with the findings of *Bouamor et al. (2018)*, indicating a high degree of overlap between the Amman and Jerusalem dialects, given their close geographical proximity.

To quantify the lexical similarity between the Irbid and Amman dialects compared to MSA, we computed the Overlap Coefficient as the percentage of lexical overlap between the *corpus*-level vocabularies of each language-dialect pair. Specifically, we collected the distinct vocabulary terms from all sentences in each dataset (MSA, Amman dialect, and Irbid dialect), and then computed the Overlap Coefficient as the ratio of shared unique words to the size of the smaller vocabulary set. This choice ensures consistent normalization across vocabulary sets of different sizes and prevents the overlap score from being biased toward larger vocabularies, following the approach adopted in prior dialectal Arabic studies such as *Bouamor et al. (2018)*. The vocabulary overlap coefficients between MSA-Irbid, MSA-Amman, and Irbid-Amman are 40.25%, 35.76%, and 57.71%, respectively. Additionally, we calculated a sentence-based pairwise overlap coefficient, which measures the ratio of shared words between two parallel sentences relative to the length of the shortest sentence, then averaged across all sentence pairs. This normalization using the shortest sentence length is adopted to fairly account for sentence length variation and avoid underestimating the overlap when shorter sentences are involved. The resulting scores for MSA-Irbid, MSA-Amman, and Irbid-Amman are 2.96%, 4.46%, and 10.24%, respectively.

The relatively low overlap scores of Irbid and Amman compared to MSA emphasize the divergence between MSA and the dialects, highlighting the need for robust translation models to facilitate communication across these forms. These overlap percentages also underscore the importance of detailed linguistic studies in gaining a deeper understanding of the subtle subregional differences within the same country. Notably, the higher overlap between the Irbid and Amman dialects compared to their overlap with MSA highlights significant similarities and shared linguistic features across dialects within the same country, suggesting a promising avenue for future investigation for more region-oriented transfer learning approaches.

**Analysis of Dictionary** As mentioned, the vocabulary overlap between MSA and Irbid sentence sets is 40.25%. This discrepancy arises partly from significant phonological changes across Arabic dialects over time, leading to distinct spellings and notable differences in pronunciation (*Maamouri et al., 2004*). Additionally, the use of synonyms,

**Table 5 Categorization of lexical divergence between Irbid dialect and MSA vocabularies in the IrbidDial dataset, including percentages for each category based on a sample of 300 (Irbid dialect)-MSA term pairs.**

| Category | Percentage | Examples: IrbidDial: MSA (English Trans.) |
|---|---|---|
| Distinct words | 57.0% | بلش:ابدأ (started ) |
| Highly divergent | 9.3% | خصيصا:خصوصي (specially) |
| Moderately divergent | 8.7% | تتسع:توسع (accommodates) |
| Slightly divergent | 25.0% | الأكمام:الكمام (sleeves) |

deviations in grammatical structure between the standard and dialectal forms of Arabic, and the substitution of multi-word phrases for single words further contribute to this divergence. Borrowed words from ancient languages that have influenced Levantine dialects, such as Aramaic, also play a role (*Neishtadt, 2015*). For example, the word زلمة in Levantine, meaning رجل in MSA ('man' in English), is originally an Aramaic term. To gain more insight into the dissimilarity between the two vocabularies, we sampled 300 term pairs from the IrbidDial-MSA dictionary and categorized each pair into one of four levels: To further analyze lexical divergence, we categorized term pairs into four classes based on Levenshtein distance, informed by a qualitative analysis of a sample of dictionary entries conducted by native speakers of both MSA and the Irbid dialect. The categories are defined as follows: (1) *Distinct Words*, where the Levenshtein distance is greater than or equal to the length of the shorter term in the pair, indicating no meaningful lexical similarity; (2) *Highly Divergent*, where the Levenshtein distance is greater than two but strictly less than the length of the shorter term, reflecting substantial but still related transformations; (3) *Moderately Divergent*, where the Levenshtein distance equals 2; and (4) *Slightly Divergent*, where the Levenshtein distance equals 1. These categories were designed to improve interpretability and highlight meaningful divergence levels based on phonological shifts and orthographic variations observed in the dialectal transformations.

Our findings, detailed in Table 5, reveal that 57% of the examined term pairs are distinct words, highlighting the potential advantage of incorporating a dictionary when prompting LLM for translation or classification tasks. Additionally, 25%, 9.3%, and 8.7% of the terms exhibit slight, moderate, and high divergence, respectively, indicating considerable variation between standard and dialectal forms even for the words that share the same or closely related roots. Based on our examination, this variation is largely attributed to differences in pronunciation, grammatical deviations due to the use of less formal structures in dialects, and other phonological shifts. These variations reflect the evolving nature of Arabic in its written form.

## PROMPTING LLM FOR DIALECTICAL MT

Prompting LLMs involves using specially formulated templates, or prompts, to adapt large, generically trained language models for specific tasks using minimal or no labeled data, facilitating zero-shot and few-shot learning. This section details the prompt templates used to translate text between the Irbid dialect and MSA. These templates were adapted and applied within a set of constraints; some were specifically tailored for dialectical

translation. We also compare these constraints and adaptations to those typically employed in MT tasks.

In our study, we employ fixed prompt templates for translation with explicit text, known as *hard prompting*. This sets our approach apart from *soft prompting*, which aims to learn embeddings (*Lester, Al-Rfou & Constant, 2021*), attention weights (*Oymak et al., 2023*), or activation functions for prompt generators (*Zhu et al., 2024*) to guide the model toward a specific task, which we leave for future exploration.

We selected GPT-4o-mini, a cost-efficient LLM introduced in July 2024, accessible through OpenAI's API. It surpasses GPT-3.5 Turbo (a commonly used LLM in research (*Brown et al., 2020*)) in various benchmarks, scoring 82% on the Massive Multitask Language Understanding (MMLU) benchmark while being 60% cheaper (*OpenAI, 2024*). In general, Generative Pre-trained Transformers (GPTs) are autoregressive LLMs with strong instruction-following capabilities, making them well-suited for prompting, particularly *prefix prompting Liu et al. (2023)*, the technique we opted for in our study. Another reason for choosing prefix prompting is its suitability for MT tasks, as it efficiently directs the language model, ensuring seamless transitions and a structured translation process *Zhang, Haddow & Birch (2023)*.

Given a pretrained and fixed LLM $\mathcal{M}$, and the testing input-output pairs $\{(x_i, y_i)\}_{i=1}^m$ where, $x_i$ is the input sentence in the source language *src*, and $y_i$ is the ground truth translation in the target language *trgt*, prompting first integrate each source test input $x_i$ into an MT prompt template $\mathcal{T}$ and then generates the translation $\hat{y}_i$ by feeding $\mathcal{M}$ with $\mathcal{T}_{x_i}$ (the prompt of $x_i$) to generate translations without performing any model fine-tuning. $\mathcal{M}$ outputs a probability distribution over possible translations of $x_i$ conditioned on the task instruction in $\mathcal{T}_{x_i}$. The desired output $\hat{y}_i$ is determined at inference time by:

$$\hat{y}_i = \underset{y}{\arg\max} \ \mathcal{M}(\mathcal{T}_{x_i}, y).$$

While ZSP relies solely on the test input $x$ to generate the translation $y$, FSP involves inputting a set of a demonstration set of $k$ parallel sentence pairs $\mathcal{D}^p = \{(x_j', y_j')\}_{j=1}^k$ concatenated into $\mathcal{T}_{x_i}$ and utilizes as few-shot context.

## Zero-shot prompting

We begin by examining a basic zero-shot slot-filling prompt template (Prompt 1), commonly used for MT (*Zhang, Haddow & Birch, 2023*; *Vilar et al., 2023*). It simply specifies the source and target language names or codes along with the source sentence without explicitly instructing the model to perform translation. In our prompting experiments, we constrained the models to output only the translation without adding any clarifications.

$$[\text{src}] : \mathbf{x_{test}}, [\text{trgt}] :. \tag{1}$$

**Identifying source language *vs.* omitting it.** In Prompt, we omitted the language name or code and relied on LM to detect the source language or dialect. The goal is to assess the

model's ability to infer the correct source language or dialect autonomously and to evaluate whether explicit language identification improves translation accuracy.

$$\mathbf{x_{test}}, [\text{trgt}] : . \tag{2}$$

**Adding a task instruction.** This prompt (Prompt 3) closely resembles Prompt 1, but it expressly starts with *'Translate the following sentence from src into trgt'* instruction. The goal is to provide clearer guidance to the model, thereby fostering more precise and context-aware translations by explicitly signaling the task.

$$\begin{aligned}&\text{Translate the following sentence from } [\text{src}] \text{ into } [\text{trgt}]\\&[\text{src}] : \mathbf{x_{test}}\\&[\text{trgt}] : .\end{aligned} \tag{3}$$

**Assigning Role to the LLM** Prompt 4 represents role-based prompting (*Zheng et al., 2024*). It builds upon Prompt 3 and assigns a specific role or persona to the model (in this case, a translator) to guide its responses and influence its behavior.

$$\begin{aligned}&\text{You are an expert translator. Your task is to translate}\\&\text{the following sentence from } [\text{src}] \text{ into } [\text{trgt}]\\&[\text{src}] : \mathbf{x_{test}}\\&[\text{trgt}] : .\end{aligned} \tag{4}$$

**Using Step-by-step Translation Prompt.** Using a step-by-step prompt, the LLM is guided systematically to translate the sentence from the source language to the target language (*Briakou et al., 2024*). This prompt views translation as a multi-turn dialogue with the LLM where each step guides the next action of the model. The core idea is to decompose the complexity of the translation process into manageable, specific instructions that the model can tackle one at a time, thereby minimizing errors and improving the overall quality of the translation.

$$\begin{aligned}&[\text{src}] : \mathbf{x_{test}}\\&\text{Step 1: Identify the key terms in the } [\text{src.}]\\&\text{Step 2: Replace them with their } [\text{trgt}] \text{ equivalents.}\\&\text{Step 3: Reconstruct the sentence in } [\text{trgt.}]\\&\text{Final Translation in } [\text{trgt}] : .\end{aligned} \tag{5}$$

**Using Back-translation Prompt.** Inspired by *Hoang et al. (2018)*, we designed an iterative back-translation prompt (Prompt 6) that instructs the model to translate a given source sentence into the target, then back-translate the resulting target sentence into the source. The model compares the back-translated sentence with the source sentence to assess semantic alignment. If a high discrepancy is detected (evaluated internally in the LLM), the MSA translation is refined, and the process is repeated iteratively. The iterative refinement continues until a maximum of $B$ iterations is reached or the source sentence translated backward matches the original input. Overall, this prompt generates several

**Table 6 List of examined constraints for consistent translations and adherence to guidelines.**

| Type | Description |
|---|---|
| Length constraints | The translation does not exceed 3 words, more or less the source than sentence. |
| Fidelity to source | The translation remains as faithful as possible to the source sentence with a literal translation. |
| Domain constraints | The translation reflects the terminology and context-specific to the travel domain. |
| Preserving named entities | The named entities, such as proper names and place names, remain unchanged. |
| Handling of transliteration | The words of foreign origin (*e.g.*, English technical terms) are translated in MSA, but retained in their transliterated form in the dialect. |
| Maintain formatting | The original formatting, including punctuation, line breaks, and numbering is preserved in the translation |
| Dialect focus constraint | Capture informal speech patterns, local expressions, and cultural aspects specific to the Irbid dialect. |
| Annotation consistency constraint | The translation avoids code-switching, represents numbers as digits, preserves punctuation and key meanings from the source text and ensures word-for-word accuracy as much as possible unless idiomatic equivalents are necessary. |

pseudo-parallel examples and outputs the target translation that achieves the best alignment between the source and target languages as the final result.

Your task is to translate a sentence from src into trgt iteratively, and
refine the translation through back−translation until the meaning is
fully preserved. Repeat the following steps:
Step 1. Translate the [src] sentence into [trgt].
Step 2. Back−translate the [trgt] sentence into [src] (Generate your
own [trgt] translation at each iteration).
Step 3. Compare the back−translated sentence to the original [scr] sentence                    (6)
Step 4. Stop after a maximum of $B$ iterations or when the back−translated
sentence closely matches the original [src] sentence.
Step 5. Return the [trgt] translation that's back−translated [scr] sentence
achieves the highest alignment in meaning to original [src] sentence
[src] : $x_{test}$
[trgt] :.

**Adding Explicit Constraints:** To gain more control over the output of the language models, we defined a list of constraints (summarized in Table 6). Some are general, while others are specifically tailored to dialectal translation. We explicitly add "constraints: [list of Constraints]" to selected prompts. This explicitly makes them act as experimental settings distinct from the main task. For example, Prompt 4 with constraints becomes:

You are an expert translator. Your task is to translate
the following sentence from [src] into [trgt]
Constraints : [List of Constraints]                    (7)
[src] : $x_{test}$
[trgt] :.

**The Language/Dialect names *vs* code.** We tried the following combinations to instantiate src and trgt in Prompt 1: *'Arabic'*, *'MSA'*, *'ar'*, and to represent the MSA side and *'Irbid Dialect'*, *'Irbid Jordanian dialect'*; *'IRB' 'Ar-IRB'* and *'Ar-jo-IRB'* to represent the Irbid city dialect. The goal is to assess the effectiveness of using abbreviations *vs.* full words in representing languages or dialects. After trying different combinations, we found that consistently using the name of the language or dialect yields superior outcomes compared to using codes, so we adopted using names over codes in all our prompts.

## Few-Shot Prompting

Few-Shot Prompting (FSP), also referred to as $k$-shot prompting (*Zhang et al., 2023*) or in-context prompting (*Brown et al., 2020*), enables the LLM to utilize $k$ exemplars (or 'shots') within the input prompt to guide the model in performing translation, thereby improving its performance. For instance, extending Prompt 4 to incorporate $k$ examplars results in the following few-shot prompt:

$$
\begin{aligned}
&\text{The following are sentences translated from [src] into [trgt]}\\
&[\text{src}] : \mathbf{x'_1}, [\text{trgt}] : \mathbf{y'_1}\\
&\qquad\qquad\vdots\\
&[\text{src}] : \mathbf{x'_k}, [\text{trgt}] : \mathbf{y'_k}\\
&\text{You are an expert translator. Your task is to similarly translate}\\
&\text{the following sentence from [src] into [trgt]}\\
&[\text{src}] : \mathbf{x_{test}}\\
&[\text{trgt}] :.
\end{aligned}
\tag{8}
$$

# EXPERIMENTAL SETTING

## Baselines

To assess the efficiency of prompting GPT-4o-mini for translation from the Irbid dialect to MSA, we thoroughly compared prompting *vs* fine-tuning on various language models. The models were fine-tuned using the Irbid dialect training set (IrbidDial-10k). We divided IrbidDial-2k into testing and development sets, each with 1,000 examples. Hyperparameters were optimized on the development set, and both fine-tuned and prompted models with optimal hyperparameter values were assessed using the testing set, from which the reported results are derived.

- **AraT5** (*Nagoudi, Elmadany & Abdul-Mageed, 2022a*) is a T5 model trained on more than 248 GB of Arabic text (70 GB of MSA and 178 GB of tweets) in both MSA and various Arabic dialects. We employed the base version, **AraT5v2-base**, along with the more recent **AraT5v2-base-1024** (*Elmadany, Nagoudi & Abdul-Mageed, 2023*), which is trained on extensive and diverse data and utilizes an extended sequence length compared to AraT5v2-base.
- **AraT5-MSAizer** (*Fares, 2024*) is an AraT5-base model fine-tuned for translating five regional Arabic dialects (Gulf, Levantine, Maghrebi, Egyptian, and Iraqi) into MSA.

- **ArabicT5** (*Alrowili & Vijay-Shanker, 2022*) an Arabic T5 model pre-trained on 17 GB of Arabic corpora, including Arabic Wikipedia and Arabic news articles.
- **mBART50** (*Liu et al., 2020*) is a multilingual encoderdecoder model primarily intended for MT. It is pre-trained by denoising full texts in 50 languages, including Arabic. Then, Multilingual Bidirectional and Auto-Regressive Transformer (mBART) is fine-tuned on parallel MT data under three settings: many-to-English, English-to-many, and many-to-many. The parallel MT data used contains 230 million parallel sentences and covers high, mid, and low-resource languages.
- **mT5** (*Xue et al., 2021*) is the multilingual adaptation of the Text-to-Text Transformer model (T5) (*Raffel et al., 2020*), which approaches the MT task as a text-to-text problem. mT5 is pre-trained on the Multilingual Colossal Clean Crawled *Corpus* (mC4), a dataset of 26.76 TB covering 101 languages, including Arabic. We fine-tuned mT5 using the IrbidDial training data.

## Evaluation metrics

We employ Bilingual Evaluation Understudy (BLEU) and Recall-Oriented Understudy for Gisting Evaluation (ROUGE), two widely used lexical-based MT evaluation metrics, alongside Crosslingual Optimized Metric for Evaluation of Translation (COMET), a model-based evaluation metric, to comprehensively assess the examined models. This combination allows for a balanced evaluation by capturing both lexical similarity and the broader quality and fidelity of translations. Notably, the inclusion of COMET is particularly valuable in the zero-shot settings of LLMs, where lexical metrics like BLEU alone may fall short in reflecting translation accuracy and contextual adequacy (*Zhang et al., 2023*). Both BLEU (*Papineni et al., 2002*) and ROUGE (*Lin, 2004*) quantify (n-gram)-based lexical overlaps with human references; however, while the BLEU score focuses on measuring the precision of how much the machine-generated translation matches reference translations, ROUGE emphasizes the recall evaluating to which extent the reference content is captured in the generated output. We report ROUGE-1, ROUGE-2, and Recall-Oriented Understudy for Gisting Evaluation–Longest Common Subsequence (ROUGE-L) F1-scores. In contrast, COMET (*Rei et al., 2020*), a (contextual model)-based metric, assesses translation quality by analyzing sentence context, human translations, and quality assessments. COMET separately encodes the source, translation and reference to obtain their sentence embeddings, then combines them to compute a quality score.

## Models selection

This section presents hyperparameter tuning applied to the models considered in this study. Early stopping (with a patience of 0) is used to tune the number of epochs for all the fine-tuned methods. We found that five epochs yielded the best BLEU scores. The T5-based models were trained with a learning rate of 5.00E−5 and a batch size of 2, constrained by resource limitations. mBART50 was trained with a learning rate of 2.00E−5 and a batch size of 8, using the default values for the remaining parameters. Regarding GPT-4o-mini, it was evaluated using direct prompting without further fine-tuning. The

**Table 7 Summary of fine-tuning evaluated on Irbid Dial test sets for translating between the Irbid dialect and MSA.**

| Approach | BLEU | COMET | ROUGE-1 | ROUGE-2 | ROUGE-L |
|---|---|---|---|---|---|
| AraT5-MSAizer | 20.72 | 0.6716 | 64.84 | 38.18 | 63.76 |
| AraT5-base | 7.74 | 0.0999 | 52.76 | 22.89 | 51.92 |
| AraT5-base-1024 | 19.94 | 0.6627 | 63.50 | 37.91 | 63.88 |
| ArabicT5 | 16.53 | 0.6205 | 61.44 | 35.20 | 61.29 |
| mT5 | 16.91 | 0.5632 | 60.67 | 34.59 | 60.09 |
| mBART50 | 15.11 | 0.6086 | 61.18 | 34.34 | 61.36 |

temperature was set to 0, which is the default setting, to ensure more deterministic and consistent responses. We found that using a temperature other than 0 degraded the performance. In FSP, the value of $k$ is tuned using the development set, and the most reasonable $k$ that balances high performance with a manageable number of samples is adopted.

# RESULTS

## Fine-tuning

This section presents the outcomes of our experiments, comparing various fine-tuning approaches. the results are summerized in Table 7.

Among the fine-tuned models, AraT5-MSAizer consistently achieves the best performance among all evaluation metrics. Its superiority over the next highest-performing model, AraT5-base-2024, underscores the value of task-specific fine-tuning on regional dialects. In contrast, AraT5-base yields lower scores (BLEU score of 7.74, COMET score of 0.0999), highlighting the limitations of models with minimal task-specific fine-tuning. AraT5-base-1024 shows a considerable improvement over AraT5-base, with a BLEU score of 19.94 and a COMET score of 0.6627, placing it as the second-best fine-tuned model, after AraT5-MSAizer. The performance gain proves the importance of training language models with diverse data. mBART50 and mT5 deliver moderate results, with BLEU scores ranging from 22.41 to 30.96, suggesting that while multilingual models are useful, they may not perform as effectively as models fine-tuned specifically for Arabic dialects, such as most AraT5 versions.

## Zero-shot prompting

This section compares the performance of the various prompt templates and constraints evaluated for ZSP in this study. Table 8 summarizes the results of the six prompts we examined, showing that prompting performance varies greatly across templates. It also demonstrates that the prompt assigning the 'You are an expert translator' role to the LLM (Prompt 4) achieved the best performance across all evaluation metrics with BLEU, COMET, and ROUGE-L scores of 27.66, 0.8277, and 60.84, respectively. Following this, Prompt 3, which provides clear task instructions, achieved the second-best performance. Prompt 2, which omits the source language, delivered the lowest performance, with BLEU, COMET, and ROUGE-L scores of 21.68, 0.3448, and 51.53, respectively.

**Table 8 Results of ZSP across various approaches with no constraints applied.** The highest scores are highlighted in bold.

| Approach | BLEU | COMET | ROUGE-1 | ROUGE-2 | ROUGE-L |
|----------|------|-------|---------|---------|---------|
| Prompt 1 | 26.39 | 0.7832 | 59.63 | 33.74 | 59.28 |
| Prompt 2 | 21.68 | 0.3448 | 51.84 | 27.64 | 51.53 |
| Prompt 3 | 27.40 | 0.8184 | 60.80 | 35.32 | 60.42 |
| Prompt 4 | **27.66** | **0.8277** | **61.21** | **35.73** | **60.84** |
| Prompt 5 | 26.69 | 0.7875 | 59.67 | 33.98 | 59.30 |
| Prompt 6 | 26.50 | 0.7815 | 59.66 | 34.13 | 59.13 |

**Note:**
The bold entries correspond to the results of the best-performing prompt.

Additionally, our analysis showed that back-translating the target sentence to the source language iteratively until reaching semantic alignment with the reference sentence does not add much improvement over even the most basic prompt (1), which is a simple slot-filling merely specifying the source sentence and requests a target translation without explicit task instructions. Further investigation into more effective ways to explicitly incorporate MT evaluation metrics (*e.g.*, BLEU) in controlling the iterative process and assessing the alignment between the source sentence and the back-translated sentences rather than relying solely on the unquantifiable judgment of the LLM.

Table 9 illustrates the performance results of applying constraints defined in this study to the best-performing prompts, namely Prompts 4 and 3, and compares them against the results obtained when no constraints are applied. We observe that except for the 'dialect focus constraint' and 'length constraint,' all other constraints resulted in comparable or slightly lower performance than the no-constraint setting. Our 'dialect focus constraint' Contributed to enhancements in BLEU by 1.40, in COMET by 2.13 percentage points, and in ROUGE-L by 0.44 for Prompt 4. For Prompt 3, the respective improvements were 1.70, 2.20 and 0.65 in BLEU, COMET and ROUGE-L, respectively.

## Few-Shot Prompting

This section presents the results of our Few-Shot Prompting experiments. We investigate the effect of the number of examples (shots), exemplar diversity, and the addition of constraints on prompting performance.

**The effect of Number of exemplars** To study the impact of the exemplars count on translation performance, we tested a wide range of $k$, including 1, 4, 7, 10, 20, 30, …, up to 150. and utilized them to perform FSP on the development set. In these experiments, Prompt 4 with the dialect focus constraint is applied to build upon the best performance of ZSP. The exemplars are sampled randomly as it has proven to reduce data biases (*Zhang et al., 2023*).

The results are as depicted in Fig. 3. It is evident that using more prompt exemplars for demonstration considerably improves translation. However, performance does not exhibit a linear growth with increasing $k$. The performance levels are off around $k = 40$, with a slight improvement in BLEU score observed as $k$ increases further. We have chosen $k = 40$

**Table 9 Results for various constraints applied to Prompt 4 and Prompt 3 in translating from Irbid dialect to MSA.** The highest scores are highlighted in bold.

| Constraint | BLEU Prompt 4 | COMET | ROUGE-L | BLEU Prompt 3 | COMET | ROUGE-L |
|---|---|---|---|---|---|---|
| No constraints. | 27.66 | 0.8277 | 60.84 | 27.40 | 0.8184 | 60.42 |
| Length constraints | 27.43 | 0.8130 | 61.05 | 27.74 | 0.8108 | 60.92 |
| Fidelity to source | 26.80 | 0.7923 | 59.74 | 26.65 | 0.7694 | 59.44 |
| Domain constraints | 27.47 | 0.8187 | 60.25 | 27.30 | 0.8218 | 60.26 |
| Preserving named entities | 25.73 | 0.7523 | 58.90 | 25.99 | 0.7442 | 58.83 |
| Handling of transliteration | 26.37 | 0.7837 | 59.53 | 26.31 | 0.7748 | 59.49 |
| Maintain formatting | 24.71 | 0.7016 | 56.37 | 24.50 | 0.6854 | 55.94 |
| Dialect focus constraint | **29.06** | **84.90** | **61.28** | **29.10** | **0.8404** | **61.07** |
| Annotation consistency const. | 25.93 | 0.7873 | 58.84 | 23.98 | 0.7596 | 58.10 |

**Note:**
The bold entries correspond to the results of the best-performing constraint.

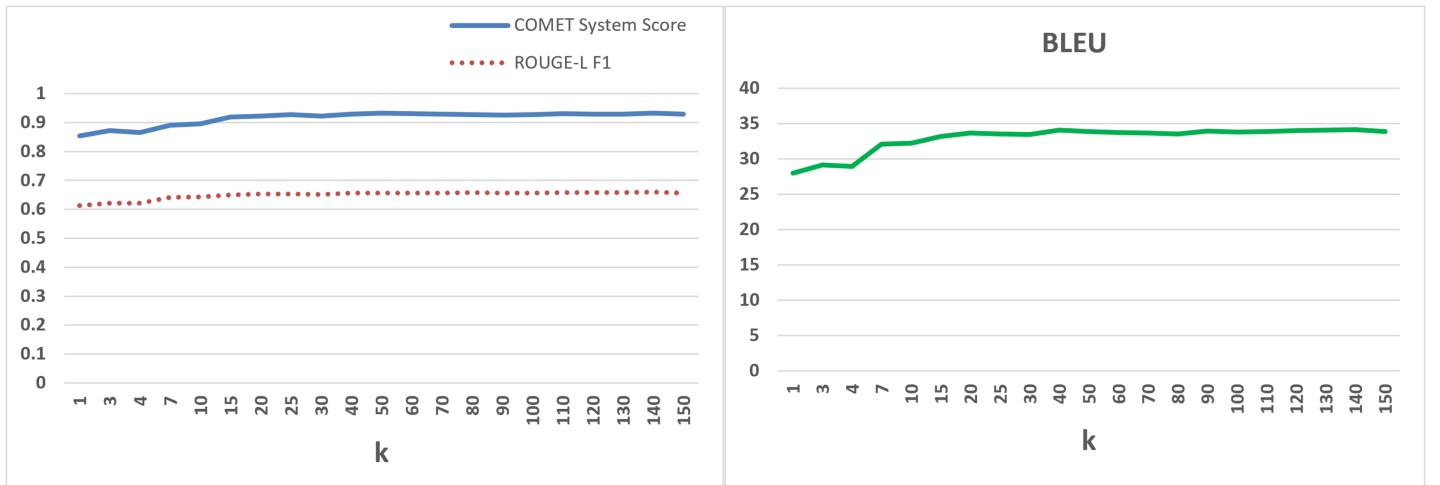

**Figure 3 Performance of FSP For various values of k using Prompt 4 and the dialect focus constraint.**

to prompt the language model to translate the test set. This decision was made despite the slight continuous improvement in BLEU score with higher $k$ values. We also observed that FSP starts to outperform ZSP from $k = 3$ on the test set, achieving a BLEU score of 28.62, a COMET score of 0.8822, and a ROUGE-L score of 62.57. Moreover, selecting higher values of $k$ has additional practical implications, such as the need for more labeled examples and increased costs associated with GPU memory usage and Application Programming Interface (API) access.

**The Effect of Adding Constraints to FSP** Table 10 presents the performance results of applying various constraints in the FSP setting, using Prompt 4 with 40 exemplars. Adding constraints generally did not yield substantial performance improvements over the no-constraint setting. The dialect focus constraint provided a slight improvement across all

**Table 10 Performance of applying various constraints in FSP.** The highest scores are highlighted in bold.

| Constraint | BLEU | COMET | ROUGE-L |
|---|---|---|---|
| No constraint | 32.51 | 0.9175 | 64.88 |
| Length constraint | 32.15 | 0.9009 | 64.80 |
| Fidelity to source | 32.59 | 0.9175 | 64.77 |
| Domain constraint | 32.12 | 0.9114 | 64.34 |
| Preserving named entities | 31.10 | 0.8721 | 64.12 |
| Handling of transliteration | 29.99 | 0.8379 | 62.98 |
| Maintain formatting | 31.33 | 0.8744 | 63.36 |
| Dialect focus constraint | **32.61** | **0.9192** | **64.94** |
| Annotation consistency constraint | 29.94 | 0.8708 | 62.98 |

**Note:**
   The bold entries correspond to the results of the best-performing constraint.

evaluation metrics, followed closely by the fidelity-to-source constraint. The dialect focus constraint consistently outperformed all other constraints as well as the no-constraint configuration, making these findings consistent with the observations from the zero-shot prompting experiments. This reinforces the effectiveness of explicitly guiding the model towards dialectal specificity when addressing dialect-to-MSA translation tasks.

**The Effect of Exemplars Diversity** To investigate whether selecting different exemplars for each value of $k$ affects the performance of FSP, we examined the values $k = 1, 3, 7, 10, 20, 30, 40$. For each $k$, we conducted three FSP runs with different random seeds. The results of these experiments, along with their averages and standard deviations, are summarized in Table 11.

We observe performance variations across all experiments for the tested $k$ values. However, these differences become negligible starting from $k = 10$, as indicated by the diminishing standard deviations. In contrast, smaller $k$ values exhibit larger standard deviations, underscoring the importance of carefully selecting $k$ to strike a balance between performance stability and resource efficiency. Choosing an appropriate $k$ is crucial for minimizing performance fluctuations while meeting reasonable resource requirements, such as API access, RAM, and GPU usage.

## Model comparison

This section presents a direct comparison between the best-performing fine-tuning approach and the best ZSP and FSP configurations. Table 12 summarizes the comparative results, highlighting the top-performing method in each evaluation metric.

FSP consistently achieves the highest performance across all evaluation metrics when compared to both fine-tuning and zero-shot prompting. It achieves a BLEU score of 32.61 and a COMET score of 0.9192, along with strong ROUGE scores, demonstrating its effectiveness in producing fluent and accurate translations. ZSP ranks second overall, reflecting the model's generalization capability when guided by optimized prompts and constraints. The superiority of FSP over ZSP highlights the benefit of augmenting prompts with examples to guide the language model towards improved performance.

**Table 11 Performance of FSP with varying numbers of examples (*k*). Results include BLEU, COMET, ROUGE-1, ROUGE-2, and ROUGE-L metrics, averaged over three runs with different random seeds.** Prompt 4 with the dialect focus constraint is applied in these experiments.

| k | Runs | BLEU | COMET | ROUGE-1 | ROUGE-2 | ROUGE-L |
|---|------|------|-------|---------|---------|---------|
| 1 | Run1 | 28.01 | 0.8543 | 0.6185 | 0.3601 | 0.6135 |
| | Run2 | 27.20 | 0.8401 | 0.6130 | 0.3529 | 0.6100 |
| | Run3 | 25.75 | 0.8040 | 0.6000 | 0.3362 | 0.5962 |
| | Avg | 26.99 | 0.8328 | 0.6105 | 0.3497 | 0.6066 |
| | SD | 1.15 | 0.0259 | 0.0095 | 0.0123 | 0.0091 |
| 3 | Run1 | 28.98 | 0.8660 | 0.6249 | 0.3683 | 0.6207 |
| | Run2 | 27.18 | 0.8408 | 0.6127 | 0.3547 | 0.6098 |
| | Run3 | 29.48 | 0.8466 | 0.6269 | 0.3732 | 0.6228 |
| | Avg | 28.55 | 0.8511 | 0.6215 | 0.3654 | 0.6178 |
| | SD | 1.21 | 0.0132 | 0.0077 | 0.0096 | 0.0070 |
| 7 | Run1 | 32.13 | 0.8903 | 0.6455 | 0.3970 | 0.6407 |
| | Run2 | 28.44 | 0.8628 | 0.6218 | 0.3657 | 0.6182 |
| | Run3 | 30.32 | 0.8589 | 0.6337 | 0.3822 | 0.6303 |
| | Avg | 30.30 | 0.8707 | 0.6337 | 0.3816 | 0.6297 |
| | SD | 1.85 | 0.0171 | 0.0119 | 0.0157 | 0.0113 |
| 10 | Run1 | 32.26 | 0.8965 | 0.6484 | 0.4004 | 0.6434 |
| | Run2 | 31.48 | 0.8881 | 0.6455 | 0.3969 | 0.6420 |
| | Run3 | 31.45 | 0.8867 | 0.6401 | 0.3928 | 0.6364 |
| | Avg | 31.73 | 0.8904 | 0.6447 | 0.3967 | 0.6406 |
| | SD | 0.46 | 0.0053 | 0.0042 | 0.0038 | 0.0037 |
| 20 | Run1 | 33.67 | 0.9228 | 0.6560 | 0.4174 | 0.6527 |
| | Run2 | 33.33 | 0.9229 | 0.6535 | 0.4159 | 0.6504 |
| | Run3 | 33.33 | 0.9232 | 0.6535 | 0.4152 | 0.6503 |
| | Avg | 33.44 | 0.9230 | 0.6543 | 0.4162 | 0.6511 |
| | SD | 0.20 | 0.0002 | 0.0014 | 0.0011 | 0.0014 |
| 30 | Run1 | 33.49 | 0.9229 | 0.6553 | 0.4152 | 0.6519 |
| | Run2 | 33.59 | 0.9232 | 0.6541 | 0.4163 | 0.6511 |
| | Run3 | 33.51 | 0.9169 | 0.6544 | 0.4160 | 0.6512 |
| | Avg | 33.53 | 0.9210 | 0.6546 | 0.4158 | 0.6514 |
| | SD | 0.05 | 0.0028 | 0.0006 | 0.0006 | 0.0004 |
| 40 | Run1 | 34.11 | 0.9291 | 0.6592 | 0.4215 | 0.6560 |
| | Run2 | 33.37 | 0.9290 | 0.6553 | 0.4171 | 0.6521 |
| | Run3 | 33.68 | 0.9187 | 0.6548 | 0.4165 | 0.6518 |
| | Avg | 33.72 | 0.9256 | 0.6564 | 0.4184 | 0.6533 |
| | SD | 0.38 | 0.0047 | 0.0020 | 0.0023 | 0.0017 |

By comparing the prompting results with fine-tuning, we observe notable discrepancies between the COMET, BLEU, and ROUGE scores. Specifically, although ZSP achieves higher COMET and BLEU scores compared to the fine-tuned AraT5-MSAizer, its ROUGE scores are lower. Similarly, while FSP shows a substantial improvement in COMET over AraT5-MSAizer, it's gain in ROUGE-L is relatively modest (1.83 points). These results

**Table 12 Comparison of the best fine-tuning method (AraT5-MSAizer) against the best ZSP and FSP configurations, evaluated on the Irbid Dialect test set for translation between Irbid dialect and MSA.** The best results are highlighted in bold.

| Approach | BLEU | COMET | ROUGE-1 | ROUGE-2 | ROUGE-L |
|---|---|---|---|---|---|
| Fine-Tuning (AraT5-MSAizer) | 20.72 | 0.6716 | 64.84 | 38.18 | 63.76 |
| ZSP (Prompt 4 + Dialect focus constraint) | 29.06 | 0.8490 | 61.66 | 36.62 | 61.28 |
| FSP (Prompt 4 + 40 exemplars + Dialect focus constraint) | **32.61** | **0.9192** | **65.30** | **40.81** | **64.94** |

Note:
The bold entries correspond to the results of the best-performing models.

suggest that fine-tuned models may better preserve lexical fidelity while prompting strategies promote more fluent, though less literal, translations. These findings underscore the need for more refined evaluation metrics that more accurately capture the nuanced trade-offs between lexical accuracy and translation quality across various training approaches.

## Discussion and limitations

This study addressed two central research questions: (1) how to build a reliable parallel dataset between MSA and a fine-grained, city-specific dialect, namely the Irbid dialect, and (2) whether prompting LLMs can effectively perform dialect-to-MSA MT, compared to the models fine-tuned using training data. To address these questions, we created the IrbidDial dataset and the Irbid-MSA dictionary. Our dataset design choices involved including sourcing MSA sentences from MADAR, selecting native speakers from diverse Irbid districts, and following clear translation guidelines. These strategies were effective in capturing the linguistic diversity of the dialect. The quality control test and the evaluation results showed high-quality translations produced by these native speakers, supporting the dataset's suitability for training and evaluating MT models focused on Jordanian dialects.

In addressing the second research question, our experiments demonstrated that prompting-based methods, especially FSP, consistently outperformed fine-tuned models across most evaluation metrics. Both ZSP and FSP surpassed the best fine-tuned model (AraT5-MSAizer) in BLEU and COMET scores, although fine-tuning remained competitive in ROUGE scores, suggesting stronger lexical fidelity. Among different prompting strategies, the combination of optimized prompts with the 'dialect focus' constraint yielded the highest performance, reconfirming the importance of explicitly guiding LLMs to handle dialectal variations. The addition of exemplars in FSP further improved performance, with performance stabilizing around 40 exemplars. Interestingly, while various constraints were explored, only the dialect focus constraint consistently improved results, confirming trends observed in ZSP experiments.

Despite these contributions, our work has several limitations. First, while IrbidDial expands dialectal MT resources, it remains limited to a single city dialect and the travel domain; broader coverage of Jordanian dialects and more domains is needed for comprehensive MT systems. Second, although prompting methods showed strong results, we did not explore more advanced prompt engineering techniques, such as soft prompting (*Lester, Al-Rfou & Constant, 2021*; *Oymak et al., 2023*), retrieval-augmented prompting

(*Feng et al., 2024*). Third, while we report consistent performance improvements, Future work should address these gaps by expanding the dataset, exploring additional dialects in Jordan for broader domains, and incorporating more sophisticated prompting strategies.

## CONCLUSION AND FUTURE WORK

Dialects, particularly city-level dialects, are often viewed as less-known varieties of languages that receive less attention compared to major dialects and national varieties. In this article, we introduce IrbidDial, a novel dataset created by translating 12,000 MSA sentences from the travel domain, originally sourced from the MADAR *corpus*, into the dialect of Irbid, a city in northern Jordan. Additionally, we developed an (Irbid dialect)-MSA dictionary with 2,155 entries to provide further resources for future research on this dataset. These resources represent the first effort to address dialectical MT for the northern Jordanian dialect, serving as an external extension to the MADAR *corpus*. We outline the data collection and annotation guidelines we established, along with the procedures and analyses employed to ensure the quality of the collected data. Furthermore, we created a set of baseline models, including prompting GPT-4o-mini, and compared them to a fine-tuning set of language models that are commonly used for MT. We defined several prompts and constraints, tailoring them to specific dialects under zero-shot and few-shot settings. Experimental results demonstrate the superiority of prompting over fine-tuning, particularly the FSP. The findings suggest that specifying roles for language models and adjusting their output to focus on the dialect and its informality, while supporting them with a sufficient number of examples, yields the best outcomes. The results also reveal the variable effectiveness of fine-tuning different language models for dialectical MT, depending on their exposure to dialectical data and the volume of data used for pretraining.

In the future, we plan to expand the *corpus* to include more under-resourced sub-dialects in Jordan. Additionally, our dialect dictionary currently provides straightforward translations without complex linguistic annotations, such as CODA or CAPHI (*Habash et al., 2018*). Future expansions are planned to include morphological and orthographical features, upgrading our dictionary to a full lexicon. Additionally, we intend to apply soft prompting techniques and compare their performance with that of the hard prompting methods discussed in this study.

### Funding

This study was funded by the Deanship of Research at Jordan University of Science and Technology under grant number 20220450. The funders had no role in study design, data collection and analysis, decision to publish, or preparation of the manuscript.

### Grant Disclosures

The following grant information was disclosed by the authors:
Deanship of Research at Jordan University of Science and Technology: 20220450.

## Competing Interests

The authors declare that they have no competing interests.

## Author Contributions

- Rasha Obeidat conceived and designed the experiments, performed the experiments, analyzed the data, performed the computation work, prepared figures and/or tables, authored or reviewed drafts of the article, and approved the final draft.
- Luay Alawneh conceived and designed the experiments, performed the experiments, authored or reviewed drafts of the article, and approved the final draft.
- Yara Al-Harahsheh analyzed the data, performed the computation work, prepared figures and/or tables, authored or reviewed drafts of the article, and approved the final draft.

## Data Availability

The code and data are available at GitHub:

- https://github.com/RashaMObeidat/IrbidDial.
- Obeidat, Rasha (2025), "IrbidDial", Mendeley Data, V2, doi: 10.17632/wd56gtw3xn.2.

## Supplemental Information

Supplemental information for this article can be found online at http://dx.doi.org/10.7717/peerj-cs.3209#supplemental-information.

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
