# Peer review of "Exploring prompting for dialectical machine translation: a focus on north Jordanian Arabic"

_PeerJ Computer Science, doi:10.7717/peerj-cs.3209_

## Round 0.1 · original submission · Major Revisions

· Academic Editor

Major Revisions

**Language Note:** When you prepare your next revision, please either (i) have a colleague who is proficient in English and familiar with the subject matter review your manuscript, or (ii) contact a professional editing service to review your manuscript. PeerJ can provide language editing services - you can contact us at [email protected] for pricing (be sure to provide your manuscript number and title). – PeerJ Staff

Reviewer 1 ·

Basic reporting

The structure of this paper is well-organized with clear statements of motivation and a background introduction. However, several improvements could be made in the following areas:

The paper contains some ambiguous statements and unclear thresholds.

1. [major] The dataset description section lacks clarity, particularly in lines 218-233. For example, the phrase "These MSA sentences and their translations" raises questions: Which translations and languages are involved? How many language pairs are there in total? Including an example dataset or a detailed table with all statistics would be helpful.

2. [major] Inconsistent terminology: Lines 264-275 confusingly interchange the terms "translators" and "annotators" while discussing the evaluation process (are reviewers also considered annotators?). The review process for these translations is also unclear—who are these reviewers and what are their qualifications?

3. [major] Missing key computational details: Line 314 lacks definitions for short, medium, and low categories. Additionally, the concept of "overlap" is not defined initially, leaving readers uncertain whether it refers to lexical, semantic, morphological, or syntactic overlap.

4. Incorrect tables: Table 2 in line 332.

5. The section "Prompting LLM for Dialectical MT" appears to be misplaced in the dataset section.

Experimental design

While prompting MT models is done in many works and is not novel at all, the specific gap is the Arabic dialects considered in this work. This is an interesting and worthwhile research gap, even though I am not very knowledgeable in this domain.

1. [major] The paper provides its own dataset, but the qualifications of translators and reviewers are unclear. Who are the translators—professional or local? What are the reviewers' qualifications?

2. [minor] While the dataset analysis is informative, some computational details are unclear. For example, the statement "The vocabulary overlap coefficients between MSA-Irbid, MSA-Amman, and Irbid Amman are 40.25%, 35.76%, and 57.71%, respectively" raises questions about the comparison methodology. How are these datasets mutually compared, and is this overlap averaged against the other datasets?

3. [major] The paper lacks motivation for certain computational choices: (a) Why use the shortest sentence length as the averaging basis? (b) Regarding lines 366-371: Are there specific reasons for the chosen lexical divergence grouping, or would a distribution plot be more effective?

4. [minor] The formulation in lines 397-404 seems irrelevant since the method does not involve any fine-tuning.

5. [minor] The training and testing set sizes (10k and 2k, respectively) should be stated at the outset. This would clarify the dataset naming conventions in lines 313-314, which currently read confusingly: "The models were 457 fine-tuned using the Irbid Dialect training set (IrbidDial-10k). We divided IrbidDial-2k into testing and 458 development sets, each with 1,000 examples."

6. [major] The paper compares fine-tuning versus prompting-based methods, experimenting with multiple prompt designs in zero-shot and few-shot settings. This raises questions about the fairness of the zero-shot versus few-shot comparison. Would adding constraints to the few-shot setting improve performance?

Validity of the findings

The results provide comprehensive multilevel comparisons, though the structure would benefit from clearer organization into distinct categories: e.g., fine-tuning versus prompting, zero-shot versus few-shot prompting, and prompt design comparisons. The current presentation packs these analyses too densely together. Moreover, while the text suggests improvements in performance “significantly”, it lacks statistical significance testing to validate these claims.

Reviewer 2 ·

Basic reporting

The manuscript is generally written in a clear way. The literature review provides a sufficient background. The structure of the article is professional, with appropriate use of figures and tables. All figure captions are fully descriptive and self-explanatory, and all tables are referenced explicitly in the text. The manuscript is mostly self-contained and aligns with the stated hypotheses. Consider adding a brief paragraph summarizing how the results address each research question explicitly. I suggest adding a discussion section, where you provide the implications and limitations of the study.

Experimental design

The study presents original primary research aligned with the journal’s aims and scope. The research question is well formulated and addresses a meaningful knowledge gap; however, further elaboration on how the study builds upon or diverges from previous work would strengthen its contribution. Add a paragraph at the end of the related work. The investigation appears technically sound and ethically conducted.

The methodology section is generally well described. To ensure full reproducibility, please consider including the code as open source. I saw the dataset, but the code will also be very helpful.

Validity of the findings

The study offers a meaningful contribution. The underlying data appear complete and appropriately handled. Conclusions are clearly articulated and remain within the scope of the presented results. Reaffirming how these findings relate to the initial research question in the discussion would enhance coherence.

·

Basic reporting

The research meets the basic requirements.

Experimental design

Yes, the research is well defined, relevant & meaningful. It is stated how research fills an identified knowledge gap.

Validity of the findings

Yes, the research meets the basic requirements.

Additional comments

Reword the introduction using standard abbreviations and remove excess filler words to improve clarity and conciseness.

The dataset is small—please clarify whether it is accredited and freely available. For example, the statement “This dataset, an extension of the MADAR multi-dialect corpus, comprises 12,000 entries translated by native speakers of the Irbid dialect” requires verification and citation.

Maintain focus on the Irbid dialect; avoid diverting attention to dialects from other countries unless directly relevant to the study’s objectives.

Clarify who the translator is - specify whether the translation was conducted by a certified institution or an individual, and provide credentials if applicable.

The description of the Irbid-to-MSA parallel dataset (e.g., “We developed the IrbidDial-MSA parallel dataset by manually translating 12,000 MSA sentences…”) must be supported with methodological transparency, including translator qualifications, translation validation procedures, and licensing status of the source data. (1000 sentences from CORPUS-25-MSA and 2000 sentences from CORPUS-5-MSA).

---

## Round 0.2 · accepted · Accept

· Academic Editor

Accept

Author has addressed the reviewer's comments properly. Thus I recommend publication of the manuscript.

Reviewer 1 ·

Basic reporting

The authors have improved the paper, taking most comments into account.

Experimental design

The authors have improved the paper.

Validity of the findings

The authors have improved the paper.

Additional comments

The authors have improved the paper. Certainly, this is not a top-notch paper, but it can be a valuable resource for the relevant subcommunity.

·

Basic reporting

'no comment'

Experimental design

'no comment'

Validity of the findings

'no comment'

Additional comments

'no comment'